# A Systematic Review and Meta-analysis Protocol on Depressive Symptoms Among Medical Students in South Asia Using Patient-reported Validated Assessment Tools: Prevalence and Associated Factors

**Mantaka Rahman**[1,2,3]*⊙, **Sharmin Sultana Tuli**[1,4]⊙, **Tamal Saha**[1,5], **Obaidullah Ibn Raquib**[2], **Afroza Tamanna Shimu**[6,7], **Saleh Mohammed Ikram**[1,8], **Emma Ashworth**[9]

1 International Centre for Diarrheal Disease Research, Bangladesh (iccddr,b), Dhaka, Bangladesh, 2 National Institute of Preventive and Social Medicine (NIPSOM), Dhaka, Bangladesh, 3 East West University (EWU), Dhaka, Bangladesh, 4 Independent University Bangladesh (IUB), Dhaka, Bangladesh, 5 American International University Bangladesh (AIUB), Dhaka, Bangladesh, 6 Dhaka Medical College and Hospital (DMCH), Dhaka, Bangladesh, 7 Green Life Medical College and Hospital (GLMCH), Dhaka, Bangladesh, 8 North South University (NSU), Dhaka, Bangladesh, 9 Liverpool John Moores University (LJMU), Liverpool, England

⊙ These authors contributed equally to this work.
* drmantaka.icddrb@gmail.com

## Abstract

Depression among medical students in South Asia is notably higher than the global average, with prevalence rates ranging from approximately 30% to 60%. Untreated depression not only affects individual student's well-being, but also impacts academic performance and future clinical competence. This study protocol aims to synthesize evidence on the prevalence and associated factors of depressive symptoms among medical students in South Asia. The study will systematically navigate *Medline (PubMed), Scopus, CINHAL, EMBASE, and APA PsycInfo* for studies available before 1st May, 2025, following PRISMA guidelines for reporting and adhering to PRISMA-P standards for protocol development. The search will search for grey literature and adopt citation chain technique, using keyword truncation and string search along with standard indexing terms. Observational longitudinal studies, including cross-sectional, cohort studies, and case-control using validated patient-reported depressive symptoms measuring tools comprising South Asian medical students. Review articles, intervention studies, case reports, case series, commentaries, preprints, conference abstracts, protocols, unpublished research, and correspondences will not be considered. No language limitation will be applied. Two independent reviewers will screen studies, with disagreements resolved by a third reviewer. The study aims to extract information on prevalence and associated factors of depressive symptoms, conducting a narrative synthesis and meta-analysis using random effect models. Forest and funnel plots will be used to visualize findings, while heterogeneity

**Data availability statement:** All study related data including search string, and relevant data are within the paper and its Supporting Information files. For any additional data, researchers can contact the corresponding author.

**Funding:** The author(s) received no specific funding for this study.

**Competing interests:** The authors have declared that no competing interests exist.

**Abbreviations:** WHO, World Health Organization; PRISMA, Preferred reporting items for systematic reviews and meta-analyses; MOOSE, Meta-analysis of observational studies in epidemiology; mNOS, modified Newcastle-Ottawa scale; PHQ-9, Patient health questionnaire-9.

will be assessed using the $I^2$ statistic, with subgroup and sensitivity analysis performed to ascertain the robustness. Risk of bias (RoB) will be measured adopting the modified Newcastle-Ottawa Scale (mNOS). Statistical analysis will be conducted using R studio v.4.3.2 and GraphPad Prism v.9.0. Understanding the prevalence and risk factors is essential to guide targeted interventions and evidence-based policy reforms that support the mental well-being of future healthcare professionals. By systematically synthesizing data from observational studies, this review will provide a comprehensive synthesis of depressive symptoms, prevalence and its correlates among medical students in South Asian region, laying the groundwork for preventive strategies and improved mental health care practices.

## Introduction

Mental health issues among medical students, particularly depression, have gained considerable attention due to their detrimental impact on academic performance, professional development, and overall well-being [1–3]. As reported by World Health Organization (WHO), depression is expected to become the leading global burden of disease by 2030, with an estimated 264 million people affected globally [4]. However, depression rates in South Asia regions, covering countries such as India, Pakistan, Bangladesh, Nepal, Sri Lanka, and Afghanistan, are notably higher than the global counterparts. Up to 40% of medical students reported depressive symptoms compared to the global average of ~27% [5,6].

Globally, medical education is known for its intense pressure, which starts from the beginning of competitive medical school admissions, long hours of study, high academic expectation, emotional strain from patient care, and the demanding nature of medical curricula. These factors, coupled with regional challenges such as inadequate availability of mental health services, self-stigma due to psychological conditions including depression, and high student-to-faculty ratios, contribute to an elevated risk of depression and anxiety [7,8]. Indeed, medical students often face higher psychological stress and burnout compared to their counterparts in other disciplines [9,10]. This is likely due to the unique stresses of their education and clinical training. Approximately 30% of medical students in Bangladesh exhibit significant depressive symptoms [11]. While its ~38% in India [12], in Pakistan ranging from 30% to 60% [13,14]. Furthermore, COVID-19 pandemic intensified the mental health challenges, with a substantial increase of depression among medical students from 35.5% to 38.7% in Nepal [15]. Evidence suggested that such issues are pervasive and not isolated to particular countries within the region. In a global context, the pooled prevalence of depression among medical students has been reported 20% in Europe, 26.8% in South America, 27% in Australia, 30.3% in North America, 31.8% in the Middle East, and 38.8% in Africa [6,16,17]. Notably, studies from South Asian countries have reported even higher rates, underscoring a pressing regional concern that surpasses the global average, including rates observed in high-income countries (HICs) like the USA and UK.

The high prevalence of anxiety and depression among medical students is also linked to social isolation, inadequate sleep, and lack of effective coping strategies globally [18]. For instance, students who employ mal-adaptive coping mechanisms, such as avoidance and denial, were more likely to experience higher levels of depression and suicidal ideation. In contrast, students who had access to support systems and employed effective coping strategies were better equipped to manage their mental health challenges [19]. In addition to the academic and cultural pressures, social media usage and problematic internet use have been identified as contributing factors to mental health issues [20]. These issues are particularly concerning, given the stigma associated with seeking professional help in many South Asian cultures [7,21]. Many medical students are reluctant to access mental health services due to fears of judgment or potential negative repercussions on their careers, leading to a cycle of untreated and potentially worsening mental health issues [1]. For instance, only 20% of medical students in Pakistan sought professional help for mental health issues due to concerns of stigma, whereas 50% of Indian and Sri-Lankan students refrained from seeking treatment [22,23].

The implications of untreated depression in medical students are profound. Beyond the personal toll on students own mental health and well-being, untreated depression among medical professionals can hinder academic success, reduce clinical competence, and ultimately affect patient care [24]. Medical students with depression are more likely to experience burnout. They experienced lower empathy, and show a decrease in their overall professional performance [19]. Thus, this issue not only impacts the students themselves but also raises serious concerns about the quality of healthcare in the future. Despite the evident need for action, mental health resources for medical students in South Asia remain inadequate. While some medical institutions have started to recognize the importance of mental health and have introduced counseling services, these initiatives are often underfunded and poorly resourced [18].

Given these consequences, it is essential to address the mental health needs of medical students through preventive measures, timely interventions, and comprehensive support systems. In addition, South Asian medical students cross-cultural diversity is different from the global community due to regional variation, environmental exposure, and socioeconomic context [25–27]. While several methodological studies have examined depression and its determinants among medical students in South Asian countries. There exists an urge of a systematic analysis to investigate the pooled prevalence of depression across the whole region, which is still lacking. Although sporadic studies exist in individual South Asian countries, the findings remain fragmented. A comprehensive systematic review and meta-analysis is needed to generate a pooled prevalence estimate focused specifically on depression using patient-reported assessment tools, account for potential regional hetero-genicity, and quantify the mental health burden, an evidence crucial for informing region-specific polices and interventions [28]. Moreover, most South Asian countries are within low-and-middle-income countries (LMICs), where limited resource and mental health infrastructure highlight the need for region-specific evidence. In addition, a broad set of associated factors exploring electronic databases, grey literature will help to identify modifiable risk factors for targeted intervention which has not systematically done yet considering the variations of depression related factors that potentially differ from other parts of the word [29–31]. Thus, this review aims to assess the pooled prevalence of depressive symptoms and it's risk factors among medical students in South Asia, with a particular emphasis on the potential consequences of untreated depression. The findings will deepen our understanding of the mental health challenges faced by medical students in South Asia, add significantly to the existing meta-analysis by providing region focused and country-specific insights into associated risk factors, and help inform targeted strategies to mitigate the impact of depression on both their academic performances and future professional lives.

## Materials and methods

### Objectives and research question

The primary objective of this study is to investigate the available literature and conduct a meta-analysis to estimate prevalence of depressive symptoms among medical students in South Asia. Additionally, the study also aims to identify and analyze the numerous pre-existing risk factors associated with depressive symptoms (refers to demographic,

academic, psycho-social, or health-related characteristics linked to depressive symptoms) in the medical student population of this region.

The research question for this study is based on the PICO framework [32], which helps structure the review by considering the Population (P), Intervention/Exposure (I), Comparison/Condition (C), and Outcome (O). The framework ensures a clear and comprehensive approach to the research topic. Here, this relates to South Asian students (undergraduate, post-graduate, and internship-level students) (P) in Medical Schools (E) experiencing depressive symptoms and its associated factors (O). There will be no comparison group (C). Based on the framework, the following research questions will guide this study "What is the prevalence of depressive symptoms and its associated factors among medical students in South Asia?". Understanding these associated factors including demographic, academic, psycho-social, institutional, and behavioural domain are crucial for designing targeted interventions and guiding policy reforms to improve the mental well-being of future healthcare professionals. By synthesizing data from various observational studies, the review seeks to offer an in-depth analysis of the mental health challenges experienced by medical students in this region.

### Study design

This systematic review and meta-analysis will follow the Preferred Reporting Items for Systematic Reviews and Meta-Analyses (PRISMA) [33] and Meta-Analysis of Observational Studies in Epidemiology (MOOSE) [34] guidelines. These guidelines ensure a robust and transparent approach in reporting, study selection, and data analysis. The entire protocol of this review will strictly adhere to PRISMA-P reporting standards (S1 Table). The systematic review will include longitudinal and cross-sectional studies, as well as cohort studies that comply with the eligibility standards. Studies will be included if they assess the prevalence of depressive symptoms using standardized, validated, patient-reported depression scales and its determinants among the South Asian medical students. The overall risk of bias (RoB) will be assessed adopting the modified NOS (mNOS) guideline [35]. The study protocol has been prospectively registered in PROSPERO [36] under registration number CRD 420251020562.

### Eligibility criteria

The inclusion and exclusion criteria are outlined as follows:

1. **Inclusion criteria.**

   - Empirical observational studies (cross-sectional, case-control, longitudinal including cohort studies) published any time before 1st May, 2025.

   - Studies that focus on medical students (undergraduate, post-graduate, internship level) from South Asia.

   - Studies reporting mental health among specific subgroups of medical students (e.g., those with long-term comorbid physical health conditions)

   - Studies that provide data on the prevalence of depression/depressive symptoms or its associated factors.

   - Studies employing patient-reported or self-administered standardized and psycho-metrically validated tools (e.g., PHQ-9, BDI/ BDI-II/SF, HADS, CES-D, ZSDS, DASS-21/ DASS-42, DSM-IV).

   - Studies published in any language.

2. **Exclusion criteria for the studies.**

   - Review articles, case reports, case series, intervention studies, study protocols, commentaries, book chapters, pre-prints, conference abstracts, or letters.

   - Studies not focused on or not specifying the exact number of medical students.

- Studies lacking quantifiable prevalence data or studies that do not provide details on the factors associated with depression/depressive symptoms.

- Unpublished and full text inaccessible studies.

- Studies that did not use validated depression assessment tools or those rating clinician-reported scales or assessment tools.

**Exclusion criteria for the population/participants.**

- Non-medical students.

- Studies reporting on medical students outside of South Asia.

- Studies on health-care or university students without any subgroup analysis of medical students.

## Information sources

The information for this review will be obtained from reputable five electronic database sources including PubMed via Medline, Scopus via Elsevier, CINHAL via EBSCOhost, EMBASE via Elsevier, APA PsycInfo via EBSCOhost. To ensure a comprehensive review, the final reference lists will be carefully searched using citation chain technique, and hand-searching to identify relevant grey literature from four sources (ProQuest, MedRxiv, Google Scholar, Government reports of South Asian countries), and regional five other online journal sources, BanglaJOL (Bangladesh), NepJOL (Nepal), PakJOL (Pakistan), SLJOL (Sri Lanka), and Shodhganga (India) as aninformation source.

## Search strategy

Preliminary searches will be carried out, utilizing Medical Subject Headings (MeSHs) terms to identify pertinent key-words related to the research topic through using five electronic databases PubMed, CINHAL, Scopus, EMBASE, APA PsycInfo. To ensure the comprehensiveness, in collaboration with an experienced in systematic reviewes, a detailed search strategy will be developed. The strategy will incorporate MeSHs (Medical Subject Headings) terms, relevant key-words, and a mix of meta-data and fields (e.g., titles, abstracts) to maximize retrieval accuracy, relevance of searching associated with the two central concepts of "Depression" and "South Asian medical students" to identify relevant studies. A preliminary version of the search string, specifically for PubMed, is outlined in Table 1 and other mentioned databases as a supplementary file (S2 Table).

## Condition

The condition includes depressive symptoms using patient-reported standardized and psycho-metrically validated tools aligned with DSM- IV or DSM- 5 criteria. Eligible instruments include Patient Health Questionnaire- 9, PHQ- 9 (threshold/ cut-off value ≥ 10), Beck Depression Inventory, BDI/ BDI-II (threshold/ cut-off value ≥ 14), Beck Depression Inventory – Short Form, BDI- SF (threshold/ cut-off value ≥ 8), Hospital Anxiety and Depression Scale – Depression subscale, HADS-D (threshold/ cut-off value ≥ 8), Center for Epidemiologic Studies Depression Scale, CES- D (threshold/ cut-off value ≥ 16), Zung Self-Rating Depression Scale, ZSDS (threshold/ cut-off value ≥ 50), Depression Anxiety Stress Scale, DASS-21/ DASS- 42 (threshold/ cut- off value ≥ 10) [1,28].

The associated factors related to depressive symptoms among medical students include (1) socio-demographic factors (age, gender, ethnicity, socio-economic status, marital status, relationship, financial problem, urban/rural residence, academic year, physical comorbidity, and type of educational institution); (2) psycho-social factors (history of mental illness, family history of depression, social support, stress levels, substance use including tobacco, alcohol, drugs, burnout, and sleep quality); (3) academia-related factors (academic performance, overburden with course curriculum, workload, exam

**Table 1. Search Strategy for PubMed Database (PICO Framework[a]).**

| Search Element | Search Terms and Structure |
|---|---|
| **Condition (C)** | (Depression[Title/Abstract]) OR (Depressive disorder[Title/Abstract])) OR (Mood disorder[Title/Abstract])) OR (Affective disorder[Title/Abstract]) |
| **Population (P)** | ("Medical students"[MeSH][b] OR "Intern doctors"[MeSH] OR "Undergraduate medical students"[MeSH] OR "Postgraduate medical students"[MeSH]) **AND** (South Asia[MeSH Terms]) OR (Southern Asia[MeSH Terms])) OR (South Asia*[MeSH Terms])) OR (Afghanistan*[MeSH Terms])) OR (Afghan[MeSH Terms])) OR (Bangladesh*[MeSH Terms])) OR (Bangladeshi[MeSH Terms])) OR (India*[MeSH Terms])) OR (Indian[MeSH Terms])) OR (Bhutan*[MeSH Terms])) OR (Bhutanese[MeSH Terms])) OR (Sri Lanka*[MeSH Terms])) OR (Sri Lankan[MeSH Terms])) OR (Maldives*[MeSH Terms])) OR (Maldivian[MeSH Terms])) OR (Nepal*[MeSH Terms])) OR (Nepalese[MeSH Terms])) OR (Pakistan*[MeSH Terms])) OR (Pakistani[MeSH Terms])) OR (Lanka*[MeSH Terms])) OR (Ceylon*[MeSH Terms]) |
| **Outcome (O)** | (Epidemiology[Title]) OR (Prevalence[Title])) OR (Trends[Title])) OR (Risk factors[Title])) OR (Associated factors[Title])) OR (Determinants[Title])) OR (Contributing factors[Title])) OR (Protective factors[Title])) OR (Aggravating factors[Title])) OR (Modifiable factors[Title])) OR (Non-modifiable factors[Title]) |

[a]PICO = Population (P), Intervention/Exposure (I), Comparison/Condition (C), and Outcome (O)

[b]MeSHs = Medical Subject Headings

Boolean operators (AND, OR, NOT), truncations, and filters (free-full texts, key-words) will be applied to refine the search results. All searches will be conducted without any date or language limitations to ensure, no relevant studies are missed.

stress, and clinical exposure); (4) lifestyle-related factors (physical activity, loneliness, diet, internet addiction, gambling, gaming addiction, and screen time).

## Population

The study will encompass from diverse ethnic groups and genders across South Asia, covering those pursuing degrees such as Bachelor of Medicine and Surgery (MBBS), Bachelor of Dental Surgery (BDS) who are being considered as medical graduates with a license to practice, as well as undergraduate medical students (from 1st to 5th academic year of medical curriculum), intern doctors, pre-clinical and clinical students, and both resident and non-resident trainees.

## Exposure

Being a medical student.

## Comparator

There will be no comparison group.

## Context

Investing depressive symptoms and its determinants among South Asian medical students.

## Outcome

The primary outcome would be to estimate the prevalence and the secondary outcome to narratively synthesize the determinants of depression or depressive symptoms among medical students in the South Asian region.

## Study selection team

The study selection will be conducted by two independent reviewers (TS, OIR) based on the pre-defined inclusion and exclusion criteria. Any conflict in this phase will be intervene by the third reviewer (SS), ensuring consistency and

accuracy in the selection process. Studies that meet the initial screening criteria will then undergo for next full-text screening. Potentially eligible studies will be finally included, and the articles will be cross checked independently by two authors (MR and SS). Any disagreements regarding study's eligibility will be discussed and resolved through consensus. The entire selection process will be documented, and a PRISMA flow diagram [33] will be created to summarize the quantity of eligible studies, exclusion, and inclusion at each stage.

## Data management

The EndNote TM [21.0] reference management software (Clarivate Analytics, Philadelphia, USA) [37] will be utilized for arranging the articles retrieved from the comprehensive search conducted across various databases. The search results and associated references will be compiled, and duplicate entries will be eliminated using EndNote reference management software. Any articles identified through manual searching will also be added to the EndNote library, if necessary. Once duplicates are excluded, the existing studies will be shifted to the Rayyan QCRI [38], a web-based application to assist in the primary screening of titles and abstracts, and to facilitate collaboration between independent reviewers. Entire library of included studies will be transported to the Rayyan platform for further review.

## Data extraction

Once the studies are selected, extraction of data will be undertaken by the authors (MR and SS) using a structured extraction form. This form will include key demographic characteristics (e.g., age, sex, ethnicity, academic year, academic level, institution, geographic location, familial history, marital status, socio-economic status); depression assessment details (e.g., depression screening or diagnostic tools used, cut-off score used, prevalence of depression including lifetime prevalence, overall prevalence, prevalence by subgroup, level of depression); study characteristics (e.g., author name, year of publication, year of data collection, study design, sample size, settings, geographic location). The main focus will be on extracting data related to the prevalence of depressive symptoms, including the specific patient-reported depression rating tools [39] used (e.g., PHQ- 9, BDI, HADS) and any reported levels of depression severity (e.g., mild, moderate, severe) as defined by the scoring thresholds or cut-off scores defined by each of those validated tools. In addition to prevalence data, the reviewers will extract information on the associated factors including socio-demographic factors, psycho-social factors, academia-related factors, and lifestyle-related factors (e.g., academic stress, burnout, peer pressure, financial crisis, social stigma, physical activity, substance use, smartphone addiction, sleep quality, relationship conflicts). In addition, other factors like socio-economic factors, gender, and institutional support systems will also be extracted. Data on mental health support mechanisms (e.g., availability of counseling services, peer support, wellness programs) will also be recorded to understand their role in mitigating depression among medical students. The extracted data will be entered into a Microsoft Excel spreadsheet, and a reviewer will verify the accuracy of the data entry to minimize errors. Any discrepancies in the data extraction process will be mitigated through consensus. The finalized extraction will be managed as well as analyzed using appropriate software tools to ensure a robust and transparent synthesis of the findings. The final extracted data set will be checked and analyzed using R v4.3.2 statistical software to facilitate a comprehensive synthesis of the findings.

## Statistical analysis

For the synthesis and analysis of the data, several steps will be employed to ensure comprehensive and accurate findings. Initially, the prevalence of depressive symptoms will be calculated for each study included in the review. A pooled prevalence estimate will be derived using a random-effects model to account for variability across studies [40]. In addition, for non-normally distributed data, Freeman–Tukey double arcsine transformation, will be applied to stabilize variance and reduce the effect of extreme prevalence values prior to pulling [41].

A meta-analysis will be conducted using R statistical software with the "meta" package [42]. This will provide an overall pooled estimate of depressive symptoms prevalence among medical students in South Asia. Subgroups will be categorized based on factors such as degree level (e.g., undergraduates vs. postgraduates), biological sex (e.g., male vs. female), and type of institution (e.g., public vs. private). Additionally, studies that utilize patient-reported different depression rating tools [39] (BDI, PHQ- 9, HADS, CES- D, ZSDS) will be grouped separately including different depressive symptoms category using standard criteria. Among the subgroups separated pooled prevalence estimation will be analysed using random-effects model. Subgroup differences will be investigated by *Q*-statistics. Heterogeneity within subgroups will allow for a comparison of depressive symptoms prevalence across studies using varying diagnostic methods. Each patient-reported tool will be treated as a separate subgroup, ensuring that the analysis reflects the potential impact of different diagnostic measures on the reported prevalence of depressive symptoms. Additionally, sensitivity analysis will also be done for assessing the robustness and generalizability of the results. This will help identify any outlier studies that might skew the results. To assess publication bias, funnel plots will be used for visual inspection of asymmetry, and Egger's test will be conducted to statistically evaluate potential bias, only if at least 10 studies are included, as the standard practice in publication bias assessment [43].

## Ethics statements

This systematic review will synthesize evidence from previously published research and therefore, formal ethical approval from the institutional review board (IRB) will not be required. As the review does not involve any primary data collection, ethical clearance is not applicable. The protocol for thus systematic review and meta-analysis has been prospectively registered with Prospective Register of Systematic Reviews (PROSPERO) under the registration number CRD 420251020562, ensuring methodological transparency and standard guidelines.

## Risk of bias assessment

Modified Newcastle-Ottawa scale (mNOS) [35], applicable for case-control, cohort, and cross-sectional observational studies, with a maximum score of 10-point will be utilized to find the overall quality of included studies. The mNOS tool will be appropriately used based on the study design, and the modified version will be used only for cross-sectional study. The modified version will include minor adaptations (e.g., defining adequate sample size (≥ 150), representativeness (random/ multi-institutional sampling, inclusion of both genders and academic years), and use of validated outcome assessment tools) to improve clarity and consistency, and the scoring criteria. Studies will be rated as good (≥ 7), medium (5–6), or low quality (< 5). Quality criteria include sample representativeness, size of the sample, exposure ascertainment, group comparability, outcome assessment, and the appropriateness of statistical tests. All studies regardless RoB (good, medium and poor quality studies) will be included in the final analysis and further evaluated by sensitivity and subgroup analysis based on study quality [35]. Cochran's $Q$ and $I^2$ statistic will be used to evaluate heterogeneity, with $I^2$ categorized as low heterogeneity (0 – 25%), moderate heterogeneity (25 – 75%), or high heterogeneity (> 75%). A random-effects model will be used for high heterogeneity, complemented by prediction intervals [44]. Results will be visualized using forest, and funnel plots. Independent reviewers will check the risk of bias, with any conflicts mitigated by involving another one.

## Strategy for data synthesis

Depressive symptoms among the medical students will be assessed by a comprehensive electronic search across the major electronic databases and gray literature searching technique. A preliminary search in PubMed on 1st May, 2025 identified 191 studies. Eligible articles will be extracted and further analyzed. After screening and eligibility checking, data from relevant studies will be systematically retrieved and organized for quantitative synthesis. In cases of primary outcome to synthesis data on demographics, and prevalence the effect measure will be proportion (% and n/N) along with 95% confidence intervals (CIs). In addition, possible reasons or factors related to the depression/depressive symptoms

will also be extracted during data extraction process. Regarding the synthesis of secondary outcome, data relating to associated factors containing variables will be extracted as odds ratios (ORs) or risk ratios (RRs) with 95% CIs (adjusted or un-adjusted) and corresponding *p*-values. All the extracted data on demographics, study quality assessment, and other associated factors will be tabulated in the main result. To address hetero-genicity across the studies, subgroup analysis on gender, secular trend, geographic location, academic year, region-specific stressors, screening tool used, study design will also be conducted across the studies to provide more context-specific findings. Egger's test and visual inspection of funnel plots will be used for detecting publication bias [43]. If quantitative data on reason and associated factors of depression/ depressive symptoms are extracted from the included studies further meta-analysis will be conducted. In the absence of quantitative synthesis, a narrative synthesis of key risk factors will be extracted to explore the depression-related factors.

Moreover, where essential information will be missing from the finally included studies, the review team will contact to the corresponding authors of the included paper via email to request clarification or additional data. If these efforts are unsuccessful, the incomplete data will either be excluded from the quantitative synthesis (meta-analysis) or, where appro-priate, imputation technique will be considered.

### Patients and public involvement

The present study protocol does not incorporate direct/indirect participation from any patients or public entities.

### Discussion

This systematic review will compile evidence from longitudinal and observational studies to determine the pooled preva-lence among medical students across South Asian countries. It will also provide a narrative synthesis of the risk factors [45] that contribute to depression and depressive symptoms, irrespective of institutional affiliation, region, or academic level. Quantitative data will be combined using meta-analysis to estimate the pooled prevalence rate, with subgroup and sensitivity analysis conducted according to standard guidelines for each approach. The review will include all published studies to date that report depression in medical students using specified diagnostic tools and eligibility.

Beyond establishing the overall prevalence, the review aims to uncover underlying causes of depression and depres-sive symptoms and identify gaps for future research and planning. However, the global prevalence of depression among medical students was reported 24.2% to 32.1% affecting one-third of medical students globally [6]. The prevalence was reported highest in the Middle East and Africa (61.1%), followed by the Americas (56.5%), Asia (41.7%), and Europe (39.9%) [46] (Fig 1). In addition, 60.9% medical students reside in LMICs [47] including South Asia. Similarly, studies had also identified that females were more vulnerable to depressive symptoms compared to male students (46.1% vs. 37.5%) and depression is also higher among pre-clinical students globally (43.7%) compared to the clinical students (41.3%) [46]. Recent studies reported, lack of infrastructure, immense workload, poor facility management, and isolation in rural health facilities discourage medical practice and may passively contribute to depression and depressive symptoms among medical students specially where resource constrain [48]. Furthermore, personal, academic, environmental, cultural, and pandemic-related stressors have been identified as key factors contributing to depression [18–20,28,30,46,49–53]. Studies reported that medical students had at least 1.65 times more likely to experience poor sleep quality, spend over 4 hours daily on smartphones, had 2.51 times higher rate of depression, and 1.75 times increased stress level compared to general students, which are the potential independent risk factors for increasing depression among themselves [49]. If similar findings align with these results, they could form the basis for larger-scale interventions and more targeted analysis of these responsible factors.

Conversely, if different patterns emerge, our review will explore potential sources of heterogeneity, such as variations in study design, methodology, and sample characteristics, which will be further examined through sub-group and sensitivity analyses. Employing robust methods to extract and synthesize data from observational studies will enable the drawing

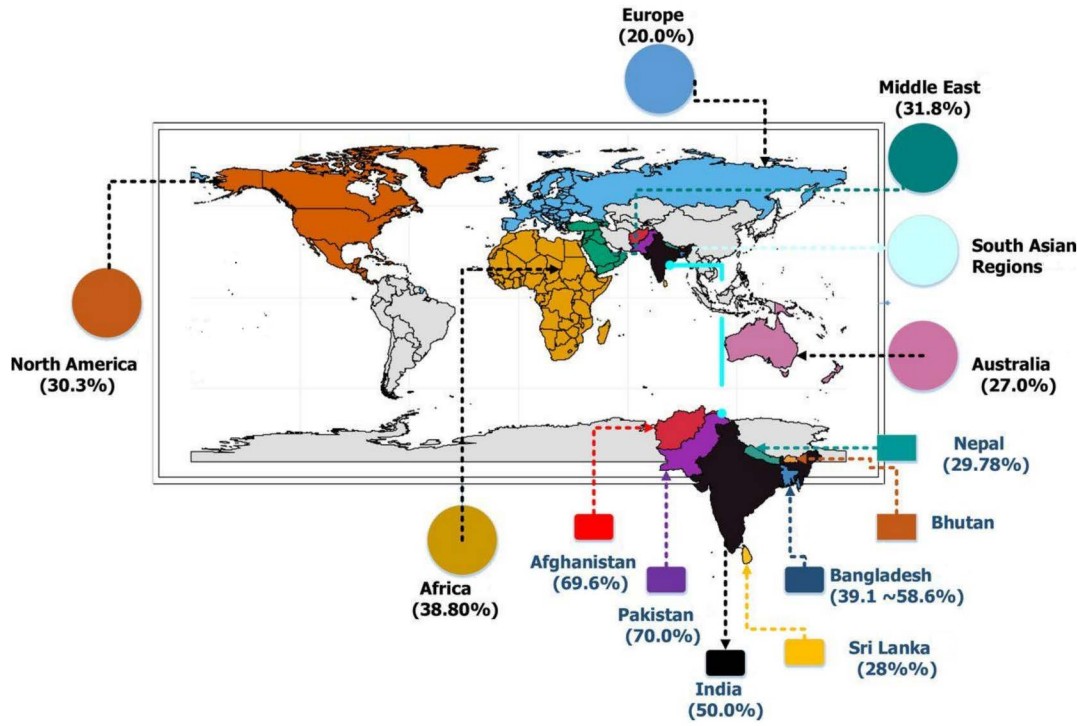

References: Puthran et al.,2016, Mekonnen et al., 2024, Bailey et al., 2018, Akhtar et al., 2020, Adhikari et al., 2021, Chomon et al., 2022, Mantaka et al., 2025, Dutta et al., 2023

Fig 1. A silent epidemic: prevalence of depression among medical students, references from global and South Asian insights.

of more definitive conclusion than any single study could provide. This study could serve as a pivotal point for designing interventions to address depression and depressive symptoms among medical students in South Asian countries.

## Strengths and limitations

The study will aim to evaluate and integrate findings from longitudinal and observational studies exploring existing literature. This study will adhere to the Cochrane Handbook [54], PRISMA- P for protocol development, PRISMA for reporting [33], MOOSE [34] and mNOS [35] guidelines for screening, reporting and quality assessment. Study screening, extraction of outcome variables, methodological, and quality evaluation will be assessed without any bias.

While heterogeneity among included studies, may pose significant challenges for meta-analysis and pooled estimates. Thus, exploring the sources of variability through subgroup analyses, ultimately strengthening the interpretation of pooled prevalence. Additionally, patient-reported depressive symptoms data may introduce methodological biases. To strengthen the reliability and generalizability of the findings, studies with significant missing data will be excluded for pooled prevalence estimation but narrative synthesis will be done to extract the associated factors if available.

## Dissemination of results and publication policy

The documents related to the study protocol is attached with the manuscript and a supplementary file. After completion of the review, a manuscript will be prepared following PRISMA guidelines [33], and will be forwarded to an international scholarly journal with the commitment to share all the relevant data. Key summary findings will be disseminated through both national,

international conferences, webinars, along with other relevant platforms. Additionally, key insights will be disseminated to policy-makers, stakeholders (e.g., including government, policy makers, medical university administration and education ministry), clinicians, general practitioners, and institutions to inform practice, guide future research, and support intervention planning.

### Potential impact

This study will generate region-specific evidence to better understand the burden and determinants of depression among medical students in South Asia. The findings of this study will inform mental health policies, promote the integration of routine mental health screening system in academic education. This will support the development of targeted prevention strategies, and guide academic institutions in implementing timely interventions to enhance student well-being and academic performance and alleviate long-term consequences of depression and depressive symptoms.

## Supporting information

**S1 Table. PRISMA-P Checklists for systematic review and meta-analysis reporting.**
(DOCX)

**S2 Table.** Search strategy for PubMed, PsycINFO, Scopus, CINAHL, EMBASE.
(DOCX)

## Acknowledgments

The authors sincerely acknowledge each affiliated author's institution, mostly icddr,b, through which the databases could be accessible.

## Authovr contributions

**Conceptualization:** Mantaka Rahman.

**Data curation:** Mantaka Rahman, Tamal Saha, Obaidullah Ibn Raquib, Afroza Tamanna Shimu, Saleh Mohammed Ikram.

**Formal analysis:** Mantaka Rahman.

**Investigation:** Mantaka Rahman.

**Methodology:** Mantaka Rahman, Sharmin Sultana.

**Resources:** Mantaka Rahman, Emma Ashworth.

**Software:** Mantaka Rahman.

**Supervision:** Mantaka Rahman, Afroza Tamanna Shimu, Emma Ashworth.

**Visualization:** Mantaka Rahman, Afroza Tamanna Shimu.

**Writing – original draft:** Mantaka Rahman, Sharmin Sultana, Afroza Tamanna Shimu.

**Writing – review & editing:** Mantaka Rahman, Tamal Saha, Obaidullah Ibn Raquib, Afroza Tamanna Shimu, Saleh Mohammed Ikram, Emma Ashworth.

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
