## [Decision Letter · Decision Letter 0]

24 Jun 2025

Dear Dr. Rahman,

Thank you for submitting your manuscript to PLOS ONE. After careful consideration, we feel that it has merit but does not fully meet PLOS ONE’s publication criteria as it currently stands. Therefore, we invite you to submit a revised version of the manuscript that addresses the points raised during the review process.

We look forward to receiving your revised manuscript.

Kind regards,

Jiawen Deng

Academic Editor

PLOS ONE

Journal Requirements:

Natural Earth (public domain): We%20note%20that%20Figure%201%20in%20your%20submission%20contain%20%5bmap/satellite%5d%20images%20which%20may%20be%20copyrighted.%20All%20PLOS%20content%20is%20published%20under%20the%20Creative%20Commons%20Attribution%20License%20(CC%20BY%204.0),%20which%20means%20that%20the%20manuscript,%20images,%20and%20Supporting%20Information%20files%20will%20be%20freely%20available%20online,%20and%20any%20third%20party%20is%20permitted%20to%20access,%20download,%20copy,%20distribute,%20and%20use%20these%20materials%20in%20any%20way,%20even%20commercially,%20with%20proper%20attribution.%20For%20these%20reasons,%20we%20cannot%20publish%20previously%20copyrighted%20maps%20or%20satellite%20images%20created%20using%20proprietary%20data,%20such%20as%20Google%20software%20(Google%20Maps,%20Street%20View,%20and%20Earth).%20For%20more%20information,%20see%20our%20copyright%20guidelines:%20 http:/journals.plos.org/plosone/s/licenses-and-copyright.%0b%0bWe%20require%20you%20to%20either%20(1)%20present%20written%20permission%20from%20the%20copyright%20holder%20to%20publish%20these%20figures%20specifically%20under%20the%20CC%20BY%204.0%20license,%20or%20(2)%20remove%20the%20figures%20from%20your%20submission:%0b%0ba.%20You%20may%20seek%20permission%20from%20the%20original%20copyright%20holder%20of%20Figure%201%20to%20publish%20the%20content%20specifically%20under%20the%20CC%20BY%204.0%20license.%20%0b%0bWe%20recommend%20that%20you%20contact%20the%20original%20copyright%20holder%20with%20the%20Content%20Permission%20Form%20(http:/journals.plos.org/plosone/s/file?id=7c09/content-permission-form.pdf)%20and%20the%20following%20text:%0b“I%20request%20permission%20for%20the%20open-access%20journal%20PLOS%20ONE%20to%20publish%20XXX%20under%20the%20Creative%20Commons%20Attribution%20License%20(CCAL)%20CC%20BY%204.0%20(http://creativecommons.org/licenses/by/4.0/).%20Please%20be%20aware%20that%20this%20license%20allows%20unrestricted%20use%20and%20distribution,%20even%20commercially,%20by%20third%20parties.%20Please%20reply%20and%20provide%20explicit%20written%20permission%20to%20publish%20XXX%20under%20a%20CC%20BY%20license%20and%20complete%20the%20attached%20form.”%0b%0bPlease%20upload%20the%20completed%20Content%20Permission%20Form%20or%20other%20proof%20of%20granted%20permissions%20as%20an%20%22Other%22%20file%20with%20your%20submission.%0b%0bIn%20the%20figure%20caption%20of%20the%20copyrighted%20figure,%20please%20include%20the%20following%20text:%20“Reprinted%20from%20%5bref%5d%20under%20a%20CC%20BY%20license,%20with%20permission%20from%20%5bname%20of%20publisher%5d,%20original%20copyright%20%5boriginal%20copyright%20year%5d.”%0b%0bb.%20If%20you%20are%20unable%20to%20obtain%20permiss http://creativecommons.org/licenses/by/4.0/).%20Please%20be%20aware%20that%20this%20license%20allows%20unrestricted%20use%20and%20distribution,%20even%20commercially,%20by%20third%20parties.%20Please%20reply%20and%20provide%20explicit%20written%20permission%20to%20publish%20XXX%20under%20a%20CC%20BY%20license%20and%20complete%20the%20attached%20form.”%0b%0bPlease%20upload%20the%20completed%20Content%20Permission%20Form%20or%20other%20proof%20of%20granted%20permissions%20as%20an%20%22Other%22%20file%20with%20your%20submission.%0b%0bIn%20the%20figure%20caption%20of%20the%20copyrighted%20figure,%20please%20include%20the%20following%20text:%20“Reprinted%20from%20%5bref%5d%20under%20a%20CC%20BY%20license,%20with%20permission%20from%20%5bname%20of%20publisher%5d,%20original%20copyright%20%5boriginal%20copyright%20year%5d.”%0b%0bb.%20If%20you%20are%20unable%20to%20obtain%20permiss" http://www.naturalearthdata.com/

Additional Editor Comments :

1. Authors' claims to be the first comprehensive synthesis are inaccurate. In fact, the authors cited several studies themselves that already pooled prevalences and risk factors across South Asian nations. Please justify the novelty and necessity of the proposed review.

2. The term "associated factors" is ill-defined. Please discuss how these factors will be categorized or specifically which factors will be assessed. Systematic reviews need to have a concrete research question; "associated factors" is too vague.

3. How exactly will the NOS be modified? Why are only medium and good quality studies included? RoB ratings can be subjective and there is a risk of leaving out relevant studies. Usually low risk studies are examined via sensitivity/subgroup analyses by RoB and rarely are they explicitly excluded.

4. How will the effects of "associated factors" be assessed? Will there be any pooling or quantitative synthesis? If so, will these be assessed as odds ratios, meta-regressions, or something else?

5. Please name specific grey literature sources that will be searched.

6. In general, there is language issues. There are long rambling sentences which are hard to read.

7. Please review your introduction and discussions, and make sure that your statements regarding mental health stigma and systemic challenges are backed by academic citations.

8. Certain parts of the manuscript seems to have been copy pasted, ?from somewhere else. The part on heterogeneity is mentioned twice in the manuscript, for instance.

9. Will any transformation be applied to the prevalence values prior to pooling? Please clarify statistical plan.

10. There is no GRADE framework available for prevalence meta-analyses. Please remove this.

11. Please remove all figures. You do not need placeholder figures nor do you need a figure for your introduction. The last figure (network plot) seems to be for illustrative purposes only and serve no real purpose.

12. You mention time cutoff for included articles being March 2025 in one section of the manuscript but May 2025 in other sections. Please explain this discrepancy.

Reviewers' comments:

Reviewer's Responses to Questions

**Comments to the Author**

1. Does the manuscript provide a valid rationale for the proposed study, with clearly identified and justified research questions?

Reviewer #1: Yes

Reviewer #2: Partly

Reviewer #3: Yes

2. Is the protocol technically sound and planned in a manner that will lead to a meaningful outcome and allow testing the stated hypotheses?

Reviewer #1: Yes

Reviewer #2: Yes

Reviewer #3: Yes

3. Is the methodology feasible and described in sufficient detail to allow the work to be replicable?

Reviewer #1: Yes

Reviewer #2: Yes

Reviewer #3: Yes

4. Have the authors described where all data underlying the findings will be made available when the study is complete?

Reviewer #1: Yes

Reviewer #2: Yes

Reviewer #3: Yes

5. Is the manuscript presented in an intelligible fashion and written in standard English?

Reviewer #1: Yes

Reviewer #2: Yes

Reviewer #3: Yes

You may also provide optional suggestions and comments to authors that they might find helpful in planning their study.

Reviewer #1: Understanding the prevalence of depression and associated risk factors is essential to guide targeted interventions and evidence-based policy reforms that support the mental well-being of future healthcare professionals. By systematically synthesizing data from observational studies, this review will provide a comprehensive synthesis of depression prevalence and its correlates among medical students in the South Asian region, laying the groundwork for preventive strategies and improved mental health

care practices.

Reviewer #2: I would like to express my sincere appreciation for the opportunity to review this valuable and comprehensive work. The manuscript presents an insightful and systematic meta-analysis study protocol for depression among medical students in South Asia, with a strong methodological framework and a high quality of included studies.

I have some minor questions and requests:

1. The objective is clearly defined, and the author used PICO (Population, Intervention, Comparator, Outcome).

2. As a depression case, what are the standard criteria used among different studies to give a standard diagnosis of depression among medical students in the South Asian region? Should be stated.

3. The author didn’t define depression cases in different studies.

4. Regarding I² categorized as low (≤25%), moderate (50%), or high (≥75%), what about the range from 25% to 50%?

5. For depression measuring tools, what is the difference between self-reported and patient-reported?

6. Although the topic is clinically or scientifically significant, any justification for conducting this study since many other similar meta-analyses were conducted globally?

7. The study's rationale didn’t justify the need for a new review…

8. Inclusion and exclusion criteria are clearly specified.

9. The author should elaborate more in the discussion.

Reviewer #3: This protocol presents a timely and well-structured plan for a systematic review and meta-analysis addressing the prevalence and associated factors of depression among medical students in South Asia. The rationale is strong and grounded in regional disparities in mental health research, with a clear and relevant research question formulated using the PICO framework.

The proposed methodology is rigorous, employing best practices such as PRISMA, MOOSE, and NOS guidelines, and registration in PROSPERO. The inclusion of subgroup analysis and sensitivity testing further adds to the protocol’s analytical strength.

Strengths:

Comprehensive search strategy, including grey literature and citation chaining.

Transparent risk of bias assessment and use of validated scales (e.g., PHQ-9, BDI).

Use of standard software (R, meta package) and appropriate statistical models.

Clear data availability and ethics statements.

Areas for Minor Revision:

1. Timeline Clarification: The projected timeline (completion in mid-2026) seems extended relative to typical protocol expectations. Consider justifying or adjusting this to reflect a more standard 12–18 month execution window.

2. Subgroup Exclusions: Excluding medical students with comorbid physical health conditions may omit relevant populations. Please clarify the rationale or consider including such data with subgroup analysis.

3. Typographical Edits: Minor inconsistencies in English usage (e.g., “analysed” vs. “analyzed”) and repeated use of placeholders like “XXXX” in figures. A proofreading pass is recommended to ensure consistency and polish.

Overall, the protocol is well-prepared and likely to yield meaningful and actionable insights for medical educators, policymakers, and researchers across South Asia. I recommend acceptance following minor revisions.

**Do you want your identity to be public for this peer review?** For information about this choice, including consent withdrawal, please see our Privacy Policy

Reviewer #1: No

Reviewer #2: **Yes: ** Assoc. Prof. Dr. Hassanain Al-Talib

Reviewer #3: No

---

## [Author Response · Author response to Decision Letter 1]

24 Jun 2025

Dear Editor,

Jiawen Deng

PLOS One.

The authors would like to express their heartfelt gratitude for the opportunity to submit the revised version of the manuscript to all the respected reviewers and the editors. Having received the mentioned suggestions made by the editor and reviewers had helped significantly to improve the overall content at all.

Our point-by-point responses to the editor and reviewers’ comments can be found below accordingly. The suggested changes have been incorporated in the revised version of the manuscript and are attached with a track change version. We tried our best to address all the comments raised by the authors and hope that the paper is now acceptable for publication in PLOS One.

Thank you very much, and we are looking forward to hearing from you soon.

On behalf of all the authors,

Dr. Mantaka Rahman (Corresponding author)

25th June, 2025.

Dhaka, Bangladesh.

Dear Reviewers,

PLOS One.

The authors would like to express their heartfelt gratitude for the opportunity to submit the revised version of the manuscript to all the respected reviewers and the editors. Having received the mentioned suggestions made by the reviewers’ and editor, had helped significantly to improve the overall content at all.

Our point-by-point responses to the reviewers’ and editor comments can be found below accordingly. The suggested changes have been incorporated in the revised version of the manuscript and are attached with a track change version. We tried our best to address all the comments raised by the authors and hope that the paper is now acceptable for publication in PLOS One.

Thank you very much, and we are looking forward to hearing from you soon.

On behalf of all the authors,

Dr. Mantaka Rahman (Corresponding author)

25th June, 2025.

Dhaka, Bangladesh.

RESPONSES TO THE ADDITIONAL EDITOR

Additional Editors Comment:

1. Editor’s Comment: Authors' claims to be the first comprehensive synthesis are inaccurate. In fact, the authors cited several studies themselves that already pooled prevalences and risk factors across South Asian nations. Please justify the novelty and necessity of the proposed review.

Authors’ response: Thank you for the valuable insight. We acknowledge that previous reviews have pooled prevalence and associated factors of depression among South Asian medical students. However, our review adds novelty by focusing specifically on studies that use validated patient-reported assessment tools (e.g., PHQ-9, BDI, HADS), allowing for a more standardized and comparable synthesis. In addition, our aim to address gaps in recent data across countries, particularly from low- and middle-income settings. To capture recent data, explore regional heterogeneity across countries, and include above mentioned grey literature sources often excluded from prior syntheses. These enhancements are intended to strengthen the evidence synthesis and provide more actionable insights for region-specific mental health interventions in this context.

2. Editor’s Comment: The term "associated factors" is ill-defined. Please discuss how these factors will be categorized or specifically which factors will be assessed. Systematic reviews need to have a concrete research question; "associated factors" is too vague. Authors’ response: Thank you for this concern. We agree that “associated factors” requires greater specificity based on the outcome of interest of the review. In our revised manuscript, we have clarified that associated factors refer to demographics (e.g., age, gender), academic (e.g., exam stress, study hours, academic performance), psychosocial (e.g., sleep quality, social support), institutional (access to mental health service, bullying, accommodation), and behavioral (e.g., screen time, substance use). These categories were supported by existing literature and other related categories (if available) will also be used to systematically extract as the associated factors. A narrative synthesis approach will be followed to extract the factors from the included studies to answer the research question.

3. Editor’s Comment: How exactly will the NOS be modified? Why are only medium and good quality studies included? RoB ratings can be subjective and there is a risk of leaving out relevant studies. Usually low risk studies are examined via sensitivity/subgroup analyses by RoB and rarely are they explicitly excluded.

Authors’ response: Thank you for the good observation. Although we previously planned to included only medium and good quality studies which may reduce the generalizability of the study and inconsistent result. Thus, as per your valuable suggestion in the revised version we have decided to include all studies (good, medium and poor) regardless RoB and further examine via sensitivity/subgroup analyses to maintain the methodological rigor.

4. Editor’s Comment: How will the effects of "associated factors" be assessed? Will there be any pooling or quantitative synthesis? If so, will these be assessed as odds ratios, meta-regressions, or something else?

Authors’ response: Thank you for your comment. We only plan to conduct a narrative synthesis of factors to identify and summarize the associated factors of depression among medical students, as reported in the included studies. Due to expected variability in study designs, measurement tools, and reported effect sizes, a quantitative synthesis (e.g., meta-analysis) of associated factors may not be feasible (Campbell et al., 2018). For narrative synthesis, the effects of associated factors will be assessed systematically by summarizing the reported associations across reported studies.

5. Editor’s Comment: Please name specific grey literature sources that will be searched. Authors’ response: The authors plan to find specific grey literature sources including ProQuest, MedRxiv, Google Scholar, Government reports, and regional online journals such as. BanglaJOL (Bangladesh), NepJOL (Nepal), PakJOL (Pakistan), SLJOL (SriLanka), Shodhganga (India). The following gray literature sources were incorporated in the revised version.

6. Editor’s Comment: In general, there is language issues. There are long rambling sentences which are hard to read.

Authors’ response: Thank you for the observation. We apologies for the inconsistency in the language issues. None of the first authors are native English speaker. However, in the revised version we tried to accommodate the issues and long rambling sentences are made a simple one for clarity.

7. Editor’s Comment: Please review your introduction and discussions, and make sure that your statements regarding mental health stigma and systemic challenges are backed by academic citations.

Authors’ response: Thank you for the observation. We have carefully rechecked the following aspects and necessary academic citations were incorporated accordingly in the above mention sections in the revised version.

8. Editor’s Comment: Certain parts of the manuscript seems to have been copy pasted,? from somewhere else. The part on heterogeneity is mentioned twice in the manuscript, for instance.

Authors’ response: We kindly apologizes for the inconvenience. The authors tried to explain heterogeneity briefly in the second section. However, for more clarity the duplicate portion of heterogeneity is revised and corrected. The authors sincerely checked thoroughly.

9. Editor’s Comment: Will any transformation be applied to the prevalence values prior to pooling? Please clarify statistical plan.

Authors’ response: Thank you for your nice statistical query. Yes, we do have plan to apply the Freeman–Tukey double arcsine transformation to stabilize variance and reduce the impact of extreme prevalence values prior to pooling (for non-normally distributed data). After that, as mentioned, a random-effects model will be used to estimate the pooled prevalence across studies.

10. Editor’s Comment: There is no GRADE framework available for prevalence meta-analyses. Please remove this.

Authors’ response: We completely agree with your comment and the following section has been removed in the revised version.

11. Editor’s Comment: Please remove all figures. You do not need placeholder figures nor do you need a figure for your introduction. The last figure (network plot) seems to be for illustrative purposes only and serve no real purpose.

Authors’ response: Thank you for your valuable concern. We comply with your suggestions and agree that we do not need figure for the introduction section. However, the figures including the network plot were used for visualization of the process like PRISMA, screening process, and overall prevalence globally at a glance. In respect to your suggestions we have removed all the figures from the main section rather added them as a supplementary file which will be accessible for anyone.

12. Editor’s Comment: You mention time cutoff for included articles being March 2025 in one section of the manuscript but May 2025 in other sections. Please explain this discrepancy.

Authors’ response: We apologizes for the inconsistency. The cut-off timeframe was up-to 1st May 2025 and thus revised accordingly throughout the sections.

RESPONSES TO REVIEWER 1

1. Reviewer’s comment: Understanding the prevalence of depression and associated risk factors is essential to guide targeted interventions and evidence-based policy reforms that support the mental well-being of future healthcare professionals. By systematically synthesizing data from observational studies, this review will provide a comprehensive synthesis of depression prevalence and its correlates among medical students in the South Asian region, laying the groundwork for preventive strategies and improved mental healthcare practices.

Authors’ response: Thank you for your positive feedback and the authors really appreciate your valuable comments and concerns. Further comments are addressed point by point to improve the overall merit and reliability of the manuscript.

RESPONSES TO REVIEWER 2

1. Reviewer’s comment: I would like to express my sincere appreciation for the opportunity to review this valuable and comprehensive work. The manuscript presents an insightful and systematic meta-analysis study protocol for depression among medical students in South Asia, with a strong methodological framework and a high quality of included studies.

Authors’ response: Thank you for your valuable insight and further comments are addressed point by point according in the revised version of the manuscript.

2. Reviewer’s comment: The objective is clearly defined, and the author used PICO (Population, Intervention, Comparator, Outcome).

Authors’ response: We appreciate and thank for the valuable comment of the reviewer.

3. Reviewer’s comment: As a depression case, what are the standard criteria used among different studies to give a standard diagnosis of depression among medical students in the South Asian region? Should be stated.

Authors’ response: Thank you for your insightful comment. As stated in the eligibility, studies will be included if they use standardized and validated patient-reported depression assessment tools such as the PHQ-9, BDI, and HADS. However, based on cut-off scores recommended by the respective instruments (e.g., PHQ-9 ≥10 for moderate depression), based on DSM-IV diagnostic criteria, will be used for level of depression categorization. To ensure clarity and transparency, we have stated and revised the manuscript to state that variations in measurement tools and diagnostic thresholds will be examined through subgroup and sensitivity analyses.

4. Reviewer’s comment: The author didn’t define depression cases in different studies.

Authors’ response: We acknowledge the comment. As previously mentioned, the depression cased across different studies will be defined using the patient-reported standardized and psychometrically validated tools (e.g., PHQ-9, BDI, HADS) following DSM-IV criteria. This way of defining depression cases in different studies is incorporated in the revised version accordingly.

5. Reviewer’s comment: Regarding I² categorized as low (≤25%), moderate (50%), or high (≥75%), what about the range from 25% to 50%?

Authors’ response: We apologies for the inconsistence in the categorization. In the revised version we clarified the ranges accordingly and categorized as low heterogeneity (0-25%), moderate heterogeneity (25-75%), or high heterogeneity (>75%).

6. Reviewer’s comment: For depression measuring tools, what is the difference between self-reported and patient-reported?

Authors’ response: Thank you for this valuable observation. We acknowledge that the terms self-reported and patient-reported were used interchangeably in the initial draft, which may have led to some confusion. We truly appreciate some inconsistence occurred in the manuscript. However, in the revised version a single terminology, patient-reported, was used throughout the manuscript to maintain consistency.

7. Reviewer’s comment: Although the topic is clinically or scientifically significant, any justification for conducting this study since many other similar meta-analyses were conducted globally?

Authors’ response: Thank you for your comment. While several global meta-analyses have assessed depression among medical students, region-specific data is crucial to compare the vulnerability between low-and-middle-income countries (LMICs) and high-income countries (HICs), especially considering ~60.69% of medical students are from LMICs (Akhter et al., 2020). In addition, there are sporadic studies in South Asia and lack regional synthesis; therefore, generating a pooled prevalence rate and investigating related factors in this region will provide potential evidence to inform further mental health strategical planning and address the growing disease burden in this context.

8. Reviewer’s comment: The study's rationale didn’t justify the need for a new review…

Authors’ response: Thank you for the observation. Although we have clarified the rationale. However, according to the concern of the reviewer we have revised and tried to justify the need for a new review and establish the rationale of the study accordingly.

9. Reviewer’s comment: Inclusion and exclusion criteria are clearly specified.

Authors’ response: We appreciate the valuable comments and insights of the reviewer.

10. Reviewer’s comment: The author should elaborate more in the discussion.

Authors’ response: According to the reviewer suggestion the discussion section has been elaborated with necessary revenant information accordingly in the revised manuscript.

RESPONSES TO REVIEWER 3

1. Reviewer’s comment: This protocol presents a timely and well-structured plan for a systematic review and meta-analysis addressing the prevalence and associated factors of depression among medical students in South Asia. The rationale is strong and grounded in regional disparities in mental health research, with a clear and relevant research question formulated using the PICO framework.

The proposed methodology is rigorous, employing best practices such as PRISMA, MOOSE, and NOS guidelines, and registration in PROSPERO. The inclusion of subgroup analysis and sensitivity testing further adds to the protocol’s analytical strength.

Authors’ response: Thank you for your positive feedback and commenting on the overall methodology and justification of the proposed review. We appreciate all your further suggestions and addressed point by point accordingly in the revised version to improve the overall quality of the manuscript.

2. Reviewer’s comment: (Strengths)

Comprehensive search strategy, including grey literature and citation chaining. Transparent risk of bias assessment and u

---

## [Decision Letter · Decision Letter 1]

23 Jul 2025

Dear Dr. Rahman,

Thank you for submitting your manuscript to PLOS ONE. After careful consideration, we feel that it has merit but does not fully meet PLOS ONE’s publication criteria as it currently stands. Therefore, we invite you to submit a revised version of the manuscript that addresses the points raised during the review process.

We look forward to receiving your revised manuscript.

Kind regards,

Jiawen Deng

Academic Editor

PLOS ONE

Journal Requirements:

**Additional Editor Comments:**

1. It is probably inappropriate to say that the review itself will follow PRISMA-P. The review itself should follow PRISMA, but the protocol should be reported following PRISMA-P.

2. At this stage I would like for the authors to lock in their information sources. Instead of saying "from a wide range of reputable sources, INCLUDING..." Simply list all databases that will be used.

3. The condition section is still not clear. Again, please list all standardized and validated tools that you are seeking in the included studies. Please also include threshold values as these tools are validated at a certain threshold level.

For associated factors, please provide the full list instead of giving examples of each category. The review methodology must be concrete in this regard.

4. Please explain the difference between MBBS and undergraduate medical students.

5. Please avoid the term "depression" in your protocol, as validated scales (such as PHQ-9) only screens for depressive SYMPTOMS, and they do not provide a formal diagnosis of depression.

6. Please provide the full list of demographic information being extracted.

7. Authors mentioned "The main focus will be on extracting...any reported levels of depression severity (e.g., mild, moderate, severe) based on DSM-IV criteria." Please confirm that you are aiming to extract severity based on DSM instead of different threshold levels in the validated tools. It is unclear how severity can be extracted based on DSM.

8. What kind of modification is being done to NOS? Also note that different versions of the NOS exists for different study types; please specify which one will be used.

9. There is still a redundant section on "Strategy for data synthesis", which curiously contains a copy-pasted version of the database search methodology.

Please very carefully review your manuscript to ensure that these types of errors are not included.

10. Please remove the section on status and timeline of the study. This is not typically included in systematic review protocols.

11. The incorrect checklist is attached. You should attach the PRISMA-P checklist, not the PRISMA checklist (this one is reserved for your final review).

12. Please review whether your figures are necessary. Figures 1 and 2 are not needed as they contain no data. Likewise, Figure 3 does not contain any information and it is unclear what the graph represents or what readers are supposed to draw from it. Figure 4 contains a summary of data from other studies and this is not typically seen in protocol publications.

Reviewers' comments:

Reviewer's Responses to Questions

**Comments to the Author**

1. Does the manuscript provide a valid rationale for the proposed study, with clearly identified and justified research questions?

Reviewer #1: Yes

2. Is the protocol technically sound and planned in a manner that will lead to a meaningful outcome and allow testing the stated hypotheses?

Reviewer #1: Yes

3. Is the methodology feasible and described in sufficient detail to allow the work to be replicable?

Reviewer #1: Yes

4. Have the authors described where all data underlying the findings will be made available when the study is complete?

Reviewer #1: Yes

5. Is the manuscript presented in an intelligible fashion and written in standard English?

Reviewer #1: Yes

You may also provide optional suggestions and comments to authors that they might find helpful in planning their study.

Reviewer #1: I am satisfied by the revisions made to the paper.

I am satisfied by the revisions made to the paper.

**Do you want your identity to be public for this peer review?** For information about this choice, including consent withdrawal, please see our Privacy Policy

Reviewer #1: No

---

## [Author Response · Author response to Decision Letter 2]

24 Jul 2025

RESPONSES TO THE ADDITIONAL EDITOR

Additional Editor Comment:

1. Editor’s Comment: It is probably inappropriate to say that the review itself will follow PRISMA-P. The review itself should follow PRISMA, but the protocol should be reported following PRISMA-P.

Authors’ response: Thank you for your comment. We agree with your concern that the protocol for this review adheres to PRISMA-P while the main reporting of the review will be following PRISMA. The related section in abstract, study design, and strength and limitation section has been revised accordingly (Page number 2, 6,15).

2. Editor’s Comment: At this stage I would like for the authors to lock in their information sources. Instead of saying "from a wide range of reputable sources, INCLUDING..." Simply list all databases that will be used.

Authors’ response: As per the editor’s suggestion we have clearly defined and locked the number of information sources we will use. In the information sources section, we have specified both the exact number and names of the electronic database sources to for improved clarity in the revised version (Page number 7-8).

3. Editor’s Comment: The condition section is still not clear. Again, please list all standardized and validated tools that you are seeking in the included studies. Please also include threshold values as these tools are validated at a certain threshold level.

For associated factors, please provide the full list instead of giving examples of each category. The review methodology must be concrete in this regard.

Authors’ response: Thank you for your valuable suggestion. We have tried to clearly clarify the condition section by adding the threshold/cut-off value of each validated assessment tools including PHQ-9 (≥10), BDI/BDI-II (≥14), BDI-SF (≥8), HADS (≥8), CES-D (≥16), ZSDS (≥50), DASS-21/DASS-42 (≥10) which we are seeking in the included studies in the revised manuscript (Page number: 9-10).

Regarding the associated factors, we have now provided the full list of associated factors identified in the review, categorized by 1) socio-demographic, 2) psychosocial, 3) academia, and 4) lifestyle domains. The section has also been reflected in the updated version (Page number: 9-10).

4. Editor’s Comment: Please explain the difference between MBBS and undergraduate medical students.

Authors’ response: Thank you for raising the point. In our context, MBBS/BDS refers to individuals who have already completed their medical or dental degree programs and are considered medical graduates or MBBS doctors in countries such as Bangladesh, India, and Nepal. In contrast, undergraduate medical students refer to those currently enrolled in the medical program (typically from 1st to 5th academic year) who have not yet completed their degree or obtained a license to practice. For more clarity, we have revised the terminology in the manuscript accordingly (Page number: 9).

5. Editor’s Comment: Please avoid the term "depression" in your protocol, as validated scales (such as PHQ-9) only screens for depressive SYMPTOMS, and they do not provide a formal diagnosis of depression.

Authors’ response: Thank you for the clarification. We acknowledge your concern that tools like the PHQ-9, BDI screen for depressive symptoms rather than providing a clinical diagnosis. However, we used the

term "depression" in line with how it will be reported in the included studies, where it is commonly referred to elevated depressive symptoms as "depression" in many literatures (DOI: 10.1001/jama.2016.17324, DOI: 10.1371/journal.pone.0221432, DOI: 10.1186/s12888-017-1382-3). But, based on your suggestion, we have revised the entire manuscript, title, and where appropriate to clarify that our analysis reflects depressive symptoms, not clinical diagnoses of Depression.

6. Editor’s Comment: Please provide the full list of demographic information being extracted.

Authors’ response: Thank you for your insightful comment. We will extract the following demographic information from each included study (age, sex, ethnicity, academic year, academic level, institution, geographic location, familial history, marital status, socio-economic status). Additional relevant demographic variables reported in the studies will also be recorded where available. In the revised version we have clarified each details of the tentative information being extracted (Page number: 11).

7. Editor’s Comment: Authors mentioned "The main focus will be on extracting...any reported levels of depression severity (e.g., mild, moderate, severe) based on DSM-IV criteria." Please confirm that you are aiming to extract severity based on DSM instead of different threshold levels in the validated tools. It is unclear how severity can be extracted based on DSM.

Authors’ response: Thank you for raising the observation. We agree with your concern and apologize for the confusion. We confirm that we will extract depression severity based on threshold / cut-off scores of validated screening tools (e.g., PHQ-9, BDI-II, DASS-21), as reported by the original included studies in the review, not based on DSM-IV criteria. The statement has been corrected accordingly in the revised manuscript (Page number: 11).

8. Editor’s Comment: What kind of modification is being done to NOS? Also note that different versions of the NOS exists for different study types; please specify which one will be used.

Authors’ response: Thank you for your valuable comment. We will assess study quality using the Newcastle-Ottawa Scale (NOS) adapted for cross-sectional studies as the majority of included studies are expected to be observational and cross-sectional in nature. Minor modifications include defining adequate sample size (≥150), representativeness (random/multi-institutional sampling, inclusion of both genders and academic years), and use of validated outcome assessment tools. The modified tool will be provided in the appendix section of the main review and will be used based on study design. In the revised version we have clarified it properly (Page number: 12).

9. Editor’s Comment: There is still a redundant section on "Strategy for data synthesis", which curiously contains a copy-pasted version of the database search methodology.

Please very carefully review your manuscript to ensure that these types of errors are not included.

Authors’ response: We kindly apologizes for the errors. The section "Strategy for data synthesis", has been revised accordingly and the redundancy has been removed in the revised version (Page number: 13).

10. Editor’s Comment: Please remove the section on status and timeline of the study. This is not typically included in systematic review protocols.

Authors’ response: According to the suggestion of the editor, the section pertaining “The status and timeline of the study” has been removed in the revised manuscript (Page number: 13).

11. Editor’s Comment: The incorrect checklist is attached. You should attach the PRISMA-P checklist, not the PRISMA checklist (this one is reserved for your final review).

Authors’ response: We agree with your concern. In the revised version we have added the PRISMA-P checklist accordingly.

12. Editor’s Comment: Please review whether your figures are necessary. Figures 1 and 2 are not needed as they contain no data. Likewise, Figure 3 does not contain any information and it is unclear what the graph represents or what readers are supposed to draw from it. Figure 4 contains a summary of data from other studies and this is not typically seen in protocol publications.

Authors’ response: Thank you for your valuable comment. We agree with your concern and have removed Figures 1, 2, and 3 accordingly. However, we believe the global summary visualization in Figure 4 offers valuable context by providing a concise overview of global statistics. We think it can be helpful for readers. To have quick overview. Therefore, we have retained it as Figure 1 in the discussion section (Figure 1, Page number: 14), with appropriate referencing and hope your further consideration.

Thank you for your kind consideration.

RESPONSES TO REVIEWER 1

1. Reviewer’s comment: I am satisfied by the revisions made to the paper.

Authors’ response: The authors highly appreciate the positive feedback of the reviewer.

---

## [Editor Report · Decision Letter 2]

11 Aug 2025

A Systematic Review and Meta-analysis Protocol on Depressive Symptoms Among Medical Students in South Asia Using Patient-reported Validated Assessment Tools: Prevalence and Associated Factors

PONE-D-25-27075R2

Dear Dr. Rahman,

We’re pleased to inform you that your manuscript has been judged scientifically suitable for publication and will be formally accepted for publication once it meets all outstanding technical requirements.

Kind regards,

Jiawen Deng

Academic Editor

PLOS ONE
---

## [Editor Report · Acceptance letter]

PONE-D-25-27075R2

PLOS ONE

Dear Dr. Rahman,

I'm pleased to inform you that your manuscript has been deemed suitable for publication in PLOS ONE. Congratulations! Your manuscript is now being handed over to our production team.

Kind regards,

on behalf of

Dr. Jiawen Deng

Academic Editor

PLOS ONE